# Myosin with hypertrophic cardiac mutation R712L has a decreased working stroke which is rescued by omecamtiv mecarbil

Aaron Snoberger[1], Bipasha Barua[2], Jennifer L Atherton[3], Henry Shuman[1], Eva Forgacs[3], Yale E Goldman[1]*, Donald A Winkelmann[2], E Michael Ostap[1]*

[1]Pennsylvania Muscle Institute, Perelman School of Medicine, University of Pennsylvania, Philadelphia, United States; [2]Department of Pathology and Laboratory Medicine, Robert Wood Johnson Medical School, Rutgers University, Piscataway, United States; [3]Department of Physiological Sciences, Eastern Virginia Medical School, Norfolk, United States

**Abstract** Hypertrophic cardiomyopathies (HCMs) are the leading cause of acute cardiac failure in young individuals. Over 300 mutations throughout β-cardiac myosin, including in the motor domain, are associated with HCM. A β-cardiac myosin motor mutation (R712L) leads to a severe form of HCM. Actin-gliding motility of R712L-myosin is inhibited, despite near-normal ATPase kinetics. By optical trapping, the working stroke of R712L-myosin was decreased 4-fold, but actin-attachment durations were normal. A prevalent hypothesis that HCM mutants are hypercontractile is thus not universal. R712 is adjacent to the binding site of the heart failure drug omecamtiv mecarbil (OM). OM suppresses the working stroke of normal β-cardiac myosin, but remarkably, OM rescues the R712L-myosin working stroke. Using a flow chamber to interrogate a single molecule during buffer exchange, we found OM rescue to be reversible. Thus, the R712L mutation uncouples lever arm rotation from ATPase activity and this inhibition is rescued by OM.

*For correspondence:
goldmany@pennmedicine.upenn.edu (YEG);
ostap@pennmedicine.upenn.edu (EMO)

Competing interests: The authors declare that no competing interests exist.

## Introduction

Hypertrophic cardiomyopathies (HCMs) affect one in 500 individuals and are the leading cause of sudden cardiac failure in individuals under 35 years of age (*Marian and Braunwald, 2017*). The disease is characterized by left ventricular hypertrophy, cardiomyocyte disarray, and interstitial fibrosis resulting in impaired diastolic function often with preserved or enhanced systolic function. Severity of HCM varies markedly, with severe cases resulting in cardiac arrhythmias and potential for sudden cardiac death. In congenital HCM, mutations occur in more than 20 sarcomeric protein genes including MYH7 (β-cardiac myosin heavy chain), the predominant myosin isoform responsible for active contraction in human ventricles (*Marian and Braunwald, 2017*; *Spudich, 2014*).

Over 300 mutations throughout the entire coding region of MYH7 have been associated with HCM (*Homburger et al., 2016*), with many of these occurring in regions predicted to affect mechanochemical activity. Some mutations are clustered in a region of the myosin motor that interacts with the thick filament, termed the 'mesa,' that stabilizes a biochemically 'off' state (*Spudich, 2015*). A widely cited model relating myosin function to disease proposes that HCM arises from myosin mutations that enhance activity yielding hypercontractile myocytes (for review, see *Spudich, 2019*). Many MYH7 mutations examined in biophysical and physiological studies have been found to cause changes in individual kinetic steps that impact the fraction of intermediates in force-producing states. While some confer apparent hypercontractile activity, no uniform kinetic signature for HCM

has emerged from these studies (e.g., *Ujfalusi et al., 2018*; *Vera et al., 2019*; *Deacon et al., 2012*). Thus, it is not clear if all HCM mutations in the myosin motor conform to the hypercontractile hypothesis. It is therefore important to perform experiments that assess the biochemical and mechanical activities of a range of HCM mutations to understand how missense mutations affect contractile activity.

R712L is a rare missense mutation in the motor domain of β-cardiac myosin that causes HCM and is characterized by sudden cardiac death (*Sakthivel et al., 2000*). R712 forms a salt bridge with E497 that is in position to stabilize the mechanical interaction between the converter/lever arm domain and the relay helix in the motor, which couples the ATP binding site to the converter (*Figure 1A*). Disruption of this salt bridge in *Drosophila* indirect flight muscle myosin resulted in impaired myosin ATPase and motility rates, and disorganized sarcomere assembly (*Kronert et al., 2015*). Since the converter domain amplifies small conformational changes in the ATPase site into large lever arm swings, its disruption could decouple ATPase activity from the lever arm swing.

R712 is located directly adjacent to the binding site for the heart failure therapeutic drug OM (*Winkelmann et al., 2015*; *Planelles-Herrero et al., 2017*). Addition of OM to β-cardiac myosin stabilizes the pre-powerstroke state of WT-myosin (*Rohde et al., 2017*), and we previously found that OM drastically reduced myosin's working stroke for translocating actin and prolonged its actin attachment duration (*Woody et al., 2018b*). Although OM abrogates the working stroke of myosin, the prolonged attachment increases calcium sensitivity in muscle fibers via cooperative activation of the thin filament (TF) regulatory system, which activates the muscle (*Governali et al., 2020*). At sub-micromolar concentrations, it has been shown to improve cardiac output in both animal models and in human clinical trials (*Malik et al., 2011*; *Teerlink et al., 2016*; *Cleland et al., 2011*; *Teerlink, 2020*). We reasoned that the HCM mutation of R712 to a leucine (R712L), located adjacent to the drug binding site, might serve as a mechanistically informative target, and might alter the kinetics and step size of β-cardiac myosin in a similar manner to OM.

In the present work, we show that the R712L mutation in recombinant human β-cardiac myosin does not directly confer gain-of-function, but rather results in inhibited motility due to a 4-fold decrease in working stroke amplitude, while only marginally affecting actin-attachment kinetics. All-atom molecular dynamics (MD) simulations of the R712L structure suggest disruption of the R712-E497 salt bridge increases the compliance of the lever arm by reducing mechanical stability of the interaction between the converter and the relay helix in the motor domain. Thus, we propose that the reduced working stroke results from uncoupling the converter/lever arm motions from the ATPase-dependent changes in the motor domain. Surprisingly, addition of OM did not further suppress motility and the working stroke, but instead rescued activity in a concentration-dependent fashion. We designed a flow chamber that enabled exchange of buffers in real time while maintaining cross-bridge cycles with individual actomyosin pairs in 3-bead optical trap assays. These solution changes allowed us to show reversible rescue of single-mutant myosin molecules by OM.

## Results

### The β-cardiac myosin HCM mutant, R712L, has impaired actin filament motility

Human β-cardiac myosin wild type (WT-myosin) and R712L mutant (R712L-myosin) HMM constructs were expressed in C2C12 myoblasts and purified. We assessed five independent preparations of WT- and R712L-myosins and found no differences in expression levels and yield of these constructs (*Figure 1—figure supplement 1A–B*). Comparable yields of the R712L proteins suggest no difference in the stability in this culture system. We next used in vitro gliding assays to measure the ability of WT- and R712L-myosins to move actin filaments. Myosins were adsorbed to a nitrocellulose-coated glass coverslip, and the proportion of filaments that moved and their speeds were determined as a function of the myosin concentration. R712L-myosin propelled actin filaments more slowly than WT-myosin at all concentrations tested, with maximum velocities of $1.46 \pm 0.11$ μm·s$^{-1}$ for WT-myosin and $0.29 \pm 0.02$ μm·s$^{-1}$ (mean ± SD) for R712L-myosin at loading concentrations 100 μg·mL$^{-1}$ (*Figure 1B,C*, *Table 1*, and *Video 1*). Although a substantial fraction of actin filaments was immobile at any given time in the presence of R712L-myosin, nearly all filaments were motile at

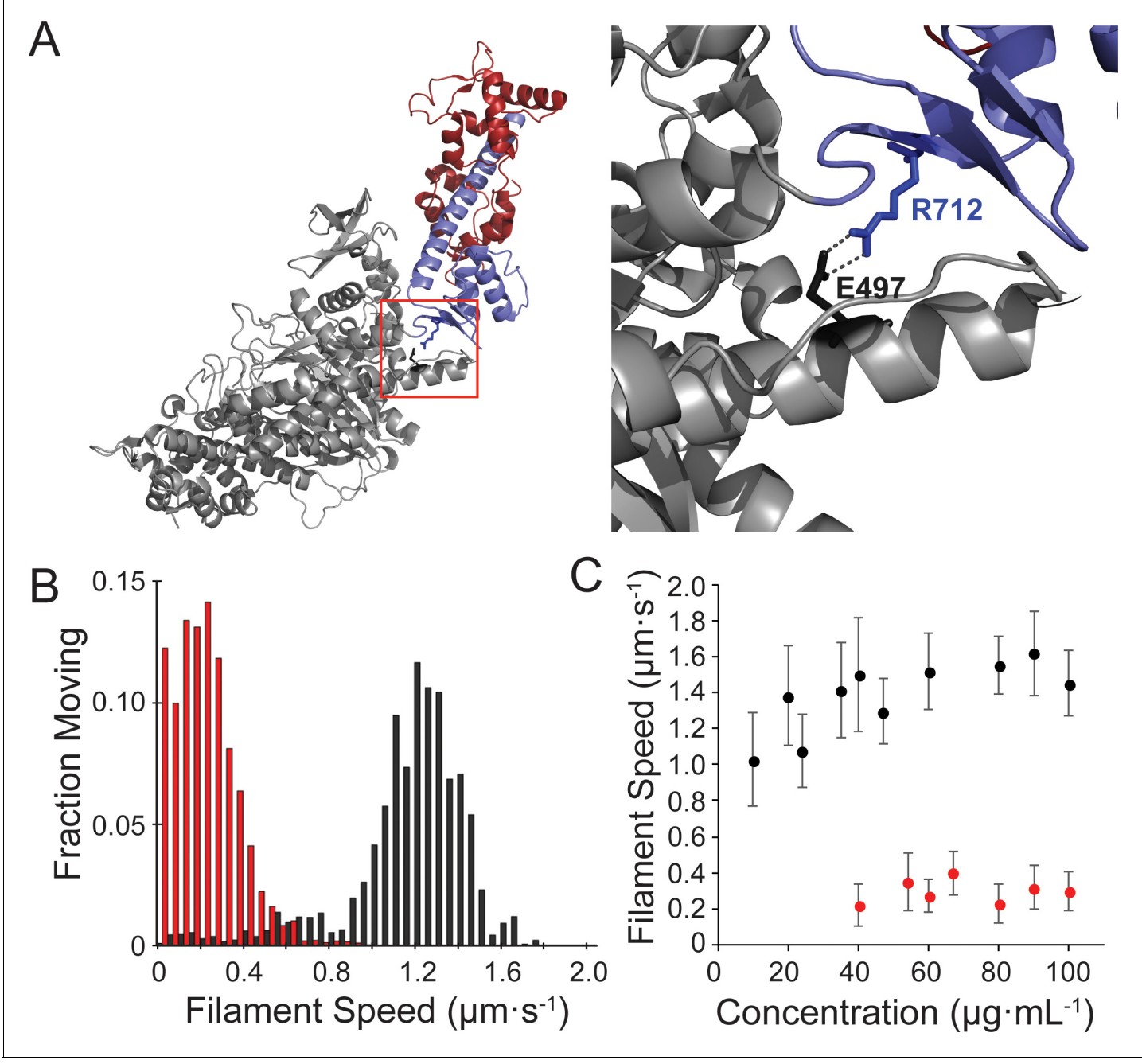

**Figure 1.** Motility of the hypertrophic cardiomyopathy mutant, R712L-myosin, is impaired. (**A**) Cartoon rendering of the β-cardiac myosin crystal structure (PDB: 5N69). The motor domain (grey, 1–707), converter/lever arm domain (blue, 708–806), and the essential light chain (red) are shown. The box indicates the region expanded to the right showing the E497-R712 salt bridge located at the fulcrum of the lever arm. (**B**) Distribution of individual filament gliding speeds from motility assays at a concentration of 100 µg·ml$^{-1}$. Wild-type-myosin (black) has a higher average motility rate compared to R712L-myosin (red). (**C**) Increasing loading concentrations of myosin were added and the average filament speed of fluorescently labelled actin filaments was assessed. Higher concentrations of R712L-myosin were required to achieve motility, and actin filaments were not observed on the surface at concentrations <40 µg·ml$^{-1}$.

The online version of this article includes the following source data and figure supplement(s) for figure 1:

**Source data 1.** Excel files with data from *Figure 1*.

**Figure supplement 1.** SDS-PAGE of WT- and R712L-myosin purification.

**Figure supplement 2.** E497D-myosin gliding filament assays.

**Figure supplement 3.** Kinetics of phosphate dissociation from thin filament (TF)-activated β-cardiac myosin.

**Figure supplement 4.** ADP dissociation from actomyosin.

*Figure 1 continued on next page*

*Figure 1 continued*

**Figure supplement 5.** ATP binding to β-cardiac myosin measured by intrinsic tryptophan fluorescence.
**Figure supplement 6.** ATP-induced dissociation of β-cardiac myosin from actin.
**Figure supplement 7.** Thin-filament (TF) activation of steady-state ATPase activity.
**Figure supplement 8.** Minimal kinetic scheme of the ATPase cycle.

some point during the assay with both constructs (*Figure 1B* and *Video 1*). Thus, the mutant motors are able to power actin gliding, but at a substantially inhibited rate.

As discussed above, the side chain of R712 forms a highly conserved salt bridge with E497 near the fulcrum of the motor's lever arm. A conserved acidic mutation (E497D) at this site also causes HCM in humans, but we found that an E497D-HMM construct powers actin gliding at nearly WT rates ($1.41 \pm 0.19\ \mu\mathrm{m \cdot s^{-1}}$; *Figure 1—figure supplement 2*). Thus, we focused our efforts on characterizing R712L.

## β-cardiac myosin R712L has normal ATPase activity and attachment durations

We studied the biochemical kinetics of WT- and R712L-myosins to determine how the mutation affects actin-activated ATPase activity (*Figure 1—figure supplements 3–8*). The R712L mutation has only minor effects on the individual rate constants of the ATPase cycle. There is a 2-fold increase in the actin-activated $P_i$ release rate (*Figure 1—figure supplement 3*) but little effect on attachment durations or ATPase rates compared to WT, indicating that phosphate release is not the rate-limiting step in the ATPase cycle of R712L-myosin. Notably, there was a ~2-fold increase in the rate of ADP release from actin-bound R712L-myosin ($142\ \mathrm{s^{-1}}$) compared to WT-myosin ($73\ \mathrm{s^{-1}}$) (*Figure 1—figure supplement 4*). This result is surprising, since normally the rate of ADP dissociation limits unloaded shortening velocity of the intact muscle, and thus a >2-fold increase in ADP release would be expected to produce a higher velocity in the gliding assay; yet, R712L-myosin has a 5-fold slower actin gliding rate (*Figure 1B,C*, and *Table 1*). These considerations suggest that ADP release may not limit actin gliding velocity for R712L-myosin, or there is a structural modification in the mutant motor that changes the linkage between the ATPase and mechanical activities.

**Table 1.** Transient and steady-state kinetic characterization of β-cardiac HMM variants with and without OM.

| Kinetic step | *Method | Rate/ equilibrium constant | WT-myosin | WT-myosin + OM | R712L-myosin | R712L-myosin + OM |
|---|---|---|---|---|---|---|
| ADP dissociation | SF-LC | $k_{-AD}$ $(s^{-1})$<br>$K_{AD}$ $(\mu M)$ | $73 \pm 2.3$<br>$19 \pm 1.2$ | $82 \pm 3.5^{\$}$<br>$22 \pm 1.1$ | $142 \pm 11^{*}$<br>$34 \pm 1.8$ | $157 \pm 6.0$<br>$36 \pm 1.8$ |
| Dissociation of AM by ATP | SF-LC | $k_{-TA}$ $(s^{-1})$ | $1191 \pm 109$ | $1129 \pm 80$ | $1201 \pm 42$ | $1168 \pm 43$ |
| ATP binding to AM | SF-LC | $k_{AT}$ $(\mu M^{-1}s^{-1})$ | $4.2 \pm 1.2$ | $5.9 \pm 1.5$ | $4.5 \pm 0.5$ | $4.4 \pm 0.5$ |
| ATP hydrolysis | SF-Fluor | $k_H + k_{-H}$ $(s^{-1})$ | $167 \pm 3.2$ | $138 \pm 3.0^{\$}$ | $87 \pm 2.0^{*}$ | $88 \pm 2.0$ |
| ATP binding | SF-Fluor | $k_T$ $(\mu M^{-1}s^{-1})$ | $5.6 \pm 0.6$ | $5.5 \pm 0.6$ | $7.1 \pm 1.0$ | $6.9 \pm 0.9$ |
| Pi release (TF) | MDCC-PBP | $k_{-DAP}$ $(s^{-1})$<br>$K_{TF}$ $(\mu M)$ | $7.3 \pm 0.8$<br><1 | $31 \pm 1.1^{\$}$<br>$3.7 \pm 0.59$ | $17 \pm 0.6^{*}$<br><1 | $17 \pm 0.4$<br>$2.1 \pm 0.78$ |
| Pi release | MDCC-PBP | $k_{-DP}$ $(s^{-1})$ | $0.014 \pm 0.002$ | $0.009 \pm 0.001^{\$}$ | $0.019 \pm 0.004$ | $0.014 \pm 0.002$ |
| TF-activated steady-state ATPase pCa = 4 | NADH-coupled assay | $V_{max}$ $(s^{-1})$<br>$K_{ATPase}$ $(\mu M)$ | $5.1 \pm 0.1$<br>$1.9 \pm 0.2$ | $2.1 \pm 0.1$<br>$0.35 \pm 0.11$ | $5.7 \pm 0.1^{*}$<br>$0.48 \pm 0.06$ | $5.5 \pm 0.1$<br>$0.43 \pm 0.06$ |
| Unloaded shortening velocity | Motility | $\mu\mathrm{m \cdot s^{-1}}$ | $1.46 \pm 0.11$ | $0.05 \pm 0.01^{\$}$ | $0.29 \pm 0.02^{*}$ | $1.1 \pm 0.26^{\#}$ |

Key kinetic rate and equilibrium constants were determined by various methods: SF-LC: stopped flow light scattering; SF-Fluor: stopped flow tryptophan fluorescence; MDCC-PBP: *N*-[2-(1-maleimidyl)ethyl]−7-(diethylamino)coumarin-3-carboxamide-phosphate binding protein; motility: in vitro motility assay; TF: native porcine thin filaments; OM: omecamtiv mecarbil. All data are presented as mean ± S.D. (N = three independent preparations). *p<0.01 (R712L-myosin- vs WT-myosin), $p<0.02 (WT-myosin + OM vs WT-myosin), #p<0.01 (R712L-myosin + OM vs R712L-myosin).

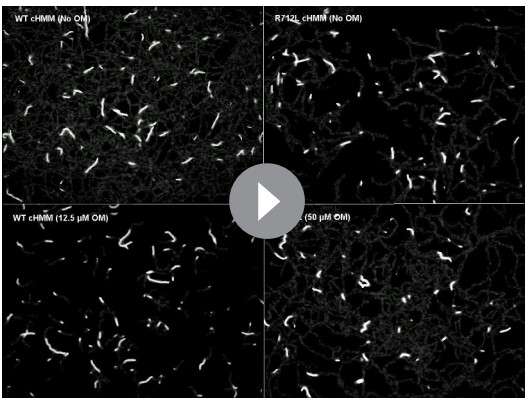

**Video 1.** The four panels of this video illustrate the contrasting effect of OM on WT- and R712L-myosins. All panels are from the same experiment with the same surface density of cardiac HMM and correspond to 100 frames captured at 5 frames/s and played back in the short clip at 30 frames/s. The faster playback frame rate is necessary to show the slow movement of the wild-type (WT)-myosin at 12.5 μM omecamtiv mecarbil (OM). Actin filament velocities measured for each condition are: WT-no OM: 1.45 μm/s, R712L-no OM: 0.27 μm/s, WT+12.5 μM: OM 0.06 μm/s, and R712L+50 μM OM: 0.81 μm/s. The last image in each panel is a maximum projection of the image stacks revealing the tracks followed by the individual filaments. Each panel is 96 x 73 μm.

https://elifesciences.org/articles/63691#video1

## R712L-myosin HMM has a defective, single-step working stroke

The conserved salt bridge involving E497-R712, which is disrupted in the mutant, is located at a mechanically crucial region that links myosin's relay helix with the converter/lever arm (*Figure 1A*; *Kronert et al., 2015*; *Winkelmann et al., 2015*; *Planelles-Herrero et al., 2017*; *Robert-Paganin et al., 2020*). We performed all-atom MD simulations to ascertain the effect of the R712L mutation on the equilibrium structure of the MgADP state of β-cardiac myosin (PDB 6FSA, *Robert-Paganin et al., 2018*). The position of the myosin lever arm and converter fluctuates during a 100 ns simulation, but remains close to the orientation found in the crystal structure (*Figure 2A*). The interface between the C-terminal end of the relay helix remains stably coupled to the β-sheet at the base of the converter domain. This coupling is mediated by stable charge interactions of R712, T761, and K762 of the converter with E497, E500, Y501 and E504 of the relay helix. Additionally, the aliphatic chain of R712 forms a hydrophobic pocket with F709, F764, E500, Y501, E504, and I506 further stabilizing the interface (*Video 2*, left). R712 was replaced with leucine in the starting structure (PDB 6FSA; *Planelles-Herrero et al., 2017*; *Robert-Paganin et al., 2018*), and a new simulation was initiated. Within 10 ns, the charge network was disrupted leaving a stable E504 to H760 and T761 backbone interaction and a transient Y501 to F709 interaction (*Video 2*, right). The relay–converter interface subsequently opened 2–3 Å, allowing water molecules into the hydrophobic pocket. Following this disruption, the lever arm rotated away from the WT orientation toward the pointed end of a hypothetical actin filament (*Figure 2A,B*, and *Video 3*). Intriguingly, these structural rearrangements result in the disruption of the binding region of the drug OM, such that the first strand of the β-sheet in the converter would sterically clash with the drug in its WT binding site (*Video 4*).

To further probe the mechanical effect of the R712L mutation, we performed steered MD simulations of the 100 ns equilibrated WT and mutant structures described above. Application of constant 70 pN force on the lever arm (see Materials and methods for details) toward the pointed end of a hypothetically bound actin filament resulted in a more substantial tilting of the mutant lever arm helix than seen for the WT (*Figure 2A,B*, and *Video 3*). Changes in the azimuthal rotation were also detected. A force higher than normally experienced by myosin (70 pN) was used to allow the simulation to proceed within an accessible computational time (*Isralewitz et al., 2001*).

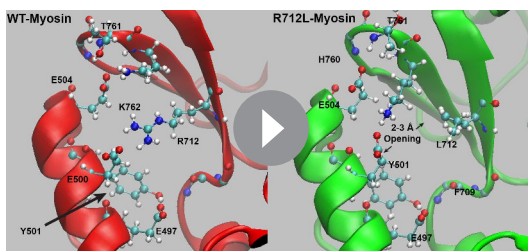

**Video 2.** Molecular dynamics simulations suggest that the R712L mutation weakens connections between the relay helix and converter domain. Simulations of (left) wild-type (WT)-myosin (6FSA with bound light chain) and (right) R712L-myosin where R712 was replaced with an L. Videos are 200 ns in length with 1 ns per frame played back at 30 frames per second.

https://elifesciences.org/articles/63691#video2

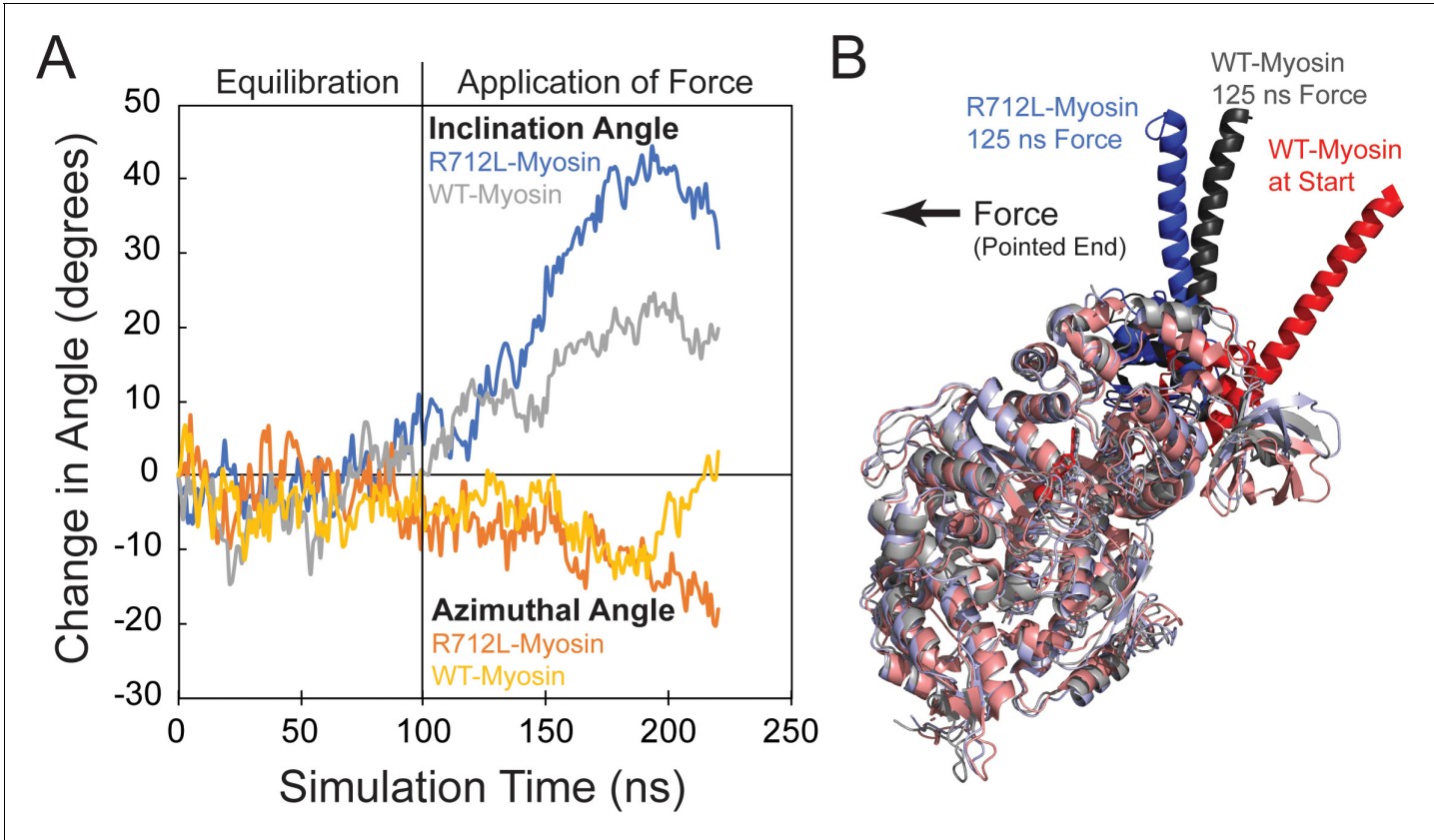

**Figure 2.** The R712L mutation affects the coupling of the motor domain to the lever arm helix. (**A**) Inclination and azimuthal angle positions of wild-type (WT)- and R712L-myosins during MD equilibration and steered molecular dynamics (MD) pulling simulations. Force is applied to the lever arm helix at 100 ns (see Materials and methods). Each curve is the average of two independent simulations. The inclination angle is defined as the angle between the actin filament axis and a vector along the lever arm helix (LAH) between residues 768–788. The change in angle is the unloaded time-zero inclination angle (127°) minus the loaded inclination angle. The azimuth is the angle between the X axis and the projection of the LAH vector onto the X-Y plane. Positive azimuth angles are counter-clockwise when viewed from the pointed end toward the barbed end of the filament. (**B**) Simulated structures after pulling the LAH (bold colors) for 125 ns. To clearly show the LAH positions, light chains were removed and motor domain residues are shown in muted colors. The arrow shows the force vector, as pulling occurred toward the pointed end and parallel to the long-axis of a hypothetically bound actin filament.

The online version of this article includes the following source data for figure 2:

**Source data 1.** Excel files with data from *Figure 2*.

Although quantitative information about the mechanical properties is not straightforwardly obtained from these simulations, the more substantial tilting of R712-myosin suggests diminished mechanical coupling of the lever arm to the motor domain in this mutant.

We hypothesized that the inhibited motility of R712L results from a decreased working stroke due to disruption of the normal mechanical integrity. To experimentally probe the effect of the R712L mutation on the working stroke, we measured the kinetics of actin attachment durations and mechanics of single myosin molecules using an optical trapping instrument that can detect sub-nanometer displacements with millisecond temporal resolution (*Woody et al., 2018b*). We used the three-bead optical trapping geometry in which a biotinylated actin filament is tethered between two laser-trapped 0.5 μm diameter polystyrene beads coated with neutravidin, creating a bead-actin-bead dumbbell (*Finer et al., 1994*; *Greenberg et al., 2017*). The dumbbell is lowered onto a larger nitrocellulose-coated pedestal bead having our HMM constructs adsorbed at low enough concentration for single molecules to interact with the filament (see Materials and methods for details). Single actomyosin binding events are detected by the decrease in covariance of the positions of the two dumbbell beads (*Figure 3A* —gray traces; see Materials and methods).

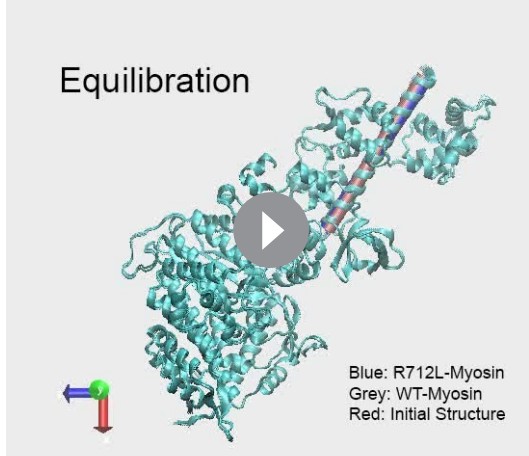

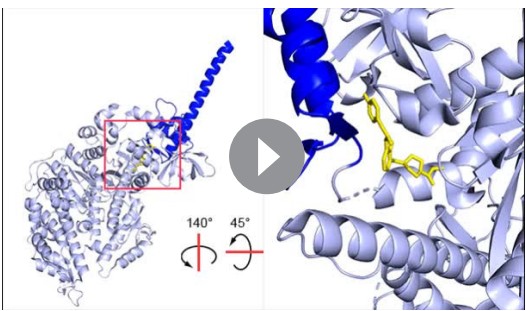

**Video 3.** The R712L mutation affects the coupling of the motor domain relay helix to the converter/lever arm. Animated molecular dynamics trajectory showing 100 ns equilibration of β-cardiac myosin (PDB: 6FSA) with (blue) and without (grey) the R712L mutation. The initial structure is shown in red, and the cylinders show the positions of the lever arm helix axes. After 100 ns, the lever arm helix is pulled for 125 ns (see Materials and methods). The arrow shows the force vector, as pulling occurred toward the pointed end and parallel to the long-axis of a hypothetically bound actin filament.
https://elifesciences.org/articles/63691#video3

**Video 4.** Morph created in PyMOL showing interpolated trajectory between the initial structure (PDB: 6FSA) and the last frame of the 100 ns equilibration (*Figure 2*, *Video 3*). The position of OM (yellow) was determined from PDB: 4PA0. The converter domain and lever arm helix are shown in dark blue. A clash (red arrow) is shown that would occur between the OM and amino acid residues P710-I713 of the converter if OM remained in its wild-type (WT) binding site. Movement of the OM to accommodate the new converter position would disrupt the extensive packing interactions between OM and the SH1-helix region and the central beta-sheet of the motor domain important for OM binding (*Winkelmann et al., 2015*); therefore, such a movement of OM is unlikely.
https://elifesciences.org/articles/63691#video4

WT-myosin actin-attachment events resulted in the decrease of the covariance signal and an observable displacement of the dumbbell due to the working stroke (*Figure 3A*, black trace). The average amplitude of the working stroke was determined by combining single-molecule interactions aligned at initial attachment times (time-forward ensemble averages) and detachment times (time-reversed averages) (*Chen et al., 2012*; Materials and methods). Time-forward ensemble averages (506 events, five molecules) in the presence of 1 μM ATP revealed an initial 3.3 nm working-stroke displacement that is considered to be associated with phosphate release, followed by an exponential rise to 4.4 nm that is consistent with a second displacement associated with ADP release (*Figure 3B*; *Chen et al., 2012*). These average displacements are similar to those reported previously for full-length and HMM constructs of β-cardiac myosin (*Woody et al., 2018b*). A single-exponential function was fit to the rising phase of the time-forward ensemble average, yielding a rate constant (99 ± 4.2 s$^{-1}$; *Table 2* and *Figure 3B*) that is similar to the biochemically measured rate of ADP release (*Figure 1—figure supplement 4*). The rate of the rising phase of the time-reversed ensemble averages leading up to detachment (4.6 ± 0.04 s$^{-1}$; *Figure 3B*) is consistent with the biochemical rate of ATP binding to nucleotide-free actomyosin at 1 μM ATP (*Figure 1—figure supplements 5–6*).

In the corresponding experiment with R712L-myosin, the data traces revealed considerably smaller displacements than observed with WT-myosin (*Figure 3A,B*, red traces). Ensemble averaging of events detected via the covariance trace (3314 events, 13 molecules) showed a drastically reduced R712L working stroke size of 1.3 nm. Not only did R712L-myosin have a small initial displacement, but unlike WT-myosin, R712L-myosin did not show a clear second step, as revealed by the similar displacements of the extension points in the time-forward and time-reversed ensemble averages. Thus, R712L does not have a detectable (<0.2 nm) second step (*Figure 3B*, red trace). Displacement only occurs promptly upon strong binding to actin.

## R712L has largely unchanged detachment rates

Using the optical trap, we measured actin attachment durations (event lifetimes) of WT- and R712L-myosins in the presence of MgATP. The distributions of actin-bound durations were adequately fitted by single exponential functions for WT-myosin (6.9 s$^{-1}$) and R712L-myosin (7.6 s$^{-1}$) in the

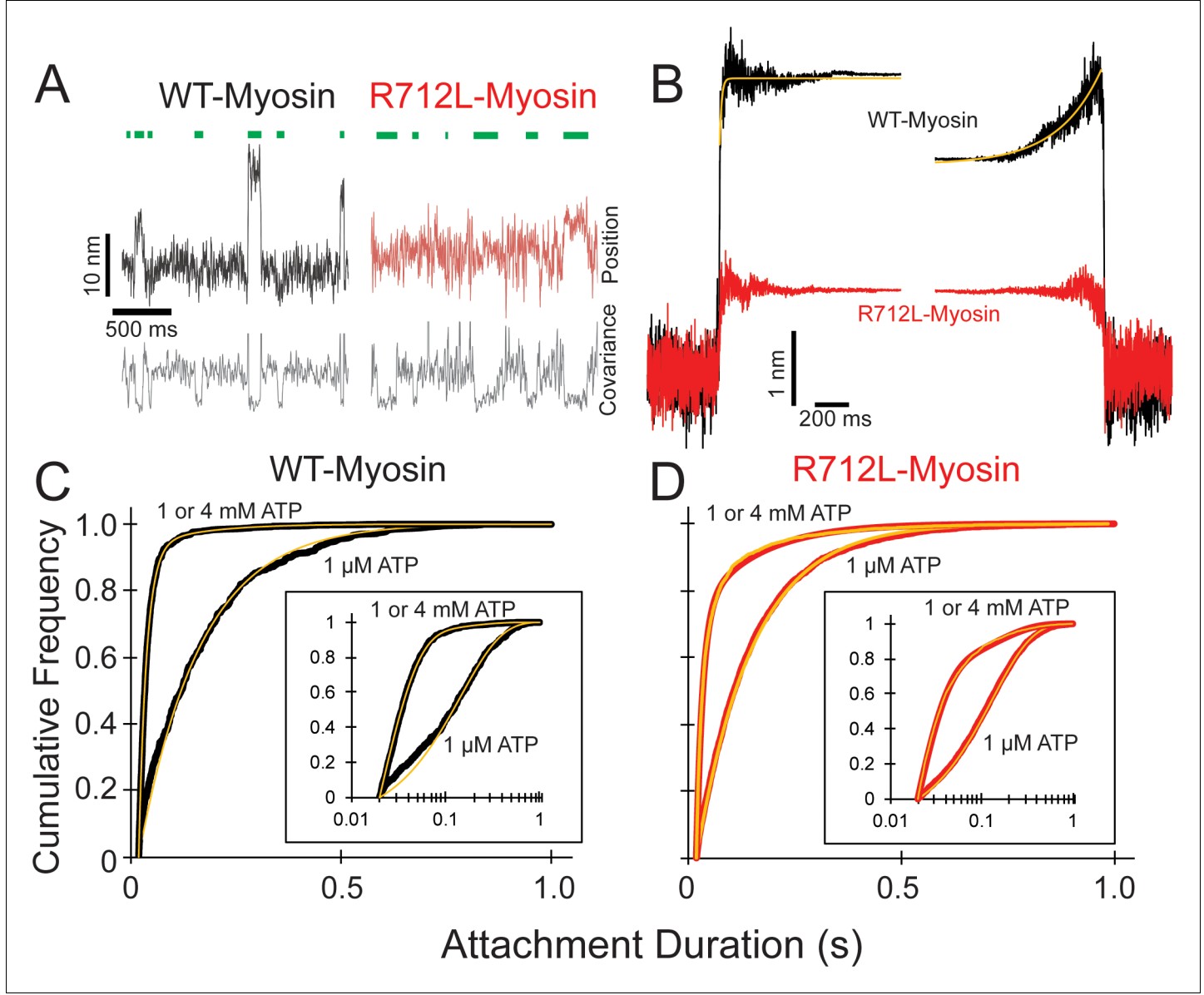

**Figure 3.** R712L-myosin has a reduced working stroke compared to WT-myosin but normal attachment durations. (A) Optical trapping displacement and covariance traces showing the position of one bead during multiple interactions of wild-type (WT)-(black) and R712L-(red)-myosins with the actin dumbbell. Green bars indicate binding events identified by decreases in bead covariance (gray traces; see Materials and methods). An averaging window of 30 ms was used for covariance traces, and the position traces shown were smoothed to 5 ms to clarify the displacements. (B) Binding events were synchronized at their beginnings and ends and were averaged forward or backward in time, respectively. The average working stroke of R712L-myosin is substantially smaller than that of WT-myosin. WT-myosin has two clear steps in its working stroke, whereas substeps could not be resolved in R712L-myosin. Yellow lines are single exponential fits to the data. (C and D) Cumulative distributions of attachment durations for WT- (C) and R712L-myosin (D) at 1 µM and saturating MgATP. Inset shows the same data on a semi-log scale. For (C) and (D) yellow lines are fitted exponential distributions, where the 1 µM ATP data were well fit by single exponentials, and the saturating 1 and 4 mM ATP data were best described by the sum of two exponentials.

The online version of this article includes the following source data for figure 3:

**Source data 1.** Excel files with data from *Figure 3*.

**Table 2.** Rates and displacement sizes of time-forward and time-reversed ensemble averages. Uncertainties are standard errors of the fit.

| | Ensemble alignment | $k_{obs}$ (s$^{-1}$) | Displacement (nm) | | |
| --- | --- | --- | --- | --- | --- |
| | | | Total | 1st substep | 2nd substep |
| Wild type | Forward | 99.1 ± 4.17 | 4.41 | 3.32 | 1.09 |
| | Reverse | 4.59 ± 0.04 | | | |
| R712L | Forward | N.D. | 1.29 | N.D. | N.D. |
| | Reverse | N.D. | | | |
| R712L + OM (50 µM) | Forward | 39.8 ± 0.91 | 2.66 | 1.82 | 0.84 |
| | Reverse | 9.61 ± 0.07 | | | |
| R712L + OM (200 µM) | Forward | 41.8 ± 0.06 | 3.42 | 2.28 | 1.14 |
| | Reverse | 6.29 ± 0.03 | | | |

[*]Data are plotted in **Figure 4C**. OM: omecamtiv mecarbil.

presence of 1 µM MgATP (**Figure 3C,D**, and **Table 3**). These rates are reasonably close to the biochemical rates (4.2 and 4.5 s$^{-1}$) expected for 1 µM MgATP binding in solution (**Figure 1—figure supplements 5–6**). At saturating MgATP (1 or 4 mM) the distributions of event lifetimes were best described by the sum of two exponentials as determined by log-likelihood ratio testing (**Figure 3C, D**, and **Table 3**; **Woody et al., 2016**). For WT-myosin, the dominant detachment rate at a high MgATP concentration (54 s$^{-1}$) is within 1.4-fold of the MgADP release rate measured in solution (73 s$^{-1}$; **Figure 1—figure supplement 4**) suggesting that this kinetic step limits actin detachment under these conditions. There was little difference in the duty ratio calculated from these attachment durations for WT-myosin (0.095) and R712L-myosin (0.078), assuming the ATPase cycle time is given by $V_{max}$. A minor component in the lifetime distribution (at 7.5 s$^{-1}$) was statistically significant, but comprised only 3% of the amplitude. The distribution of R712L-myosin actin attachment durations was also well described by the sum of two exponentials, with the predominant component (70 s$^{-1}$; **Figure 3D** and **Table 3**) similar to WT. This rate is >2-fold slower than the biochemically determined rate of ADP release in solution (142 s$^{-1}$; **Figure 1—figure supplement 5**), suggesting that a rate-limiting transition occurs before the ADP release step. The minor component had a similar rate (7.4 s$^{-1}$) as WT-myosin but comprised a larger fraction (9%) of the total (**Table 3**). Although the cause of this slow component is not known, it may be the result of enzymatically inactive motors. The increased fraction observed with R712L-myosin may explain the increased number of paused filaments observed in the motility assays (**Figure 1B**).

## Omecamtiv mecarbil rescues the motility of the R712L by restoring its working stroke

Crystal structures of myosin (**Winkelmann et al., 2015**; **Planelles-Herrero et al., 2017**) reveal that R712L is located near the binding pocket for OM, a drug in phase-3 clinical trials that increases cardiac ejection fraction in HCM patients (**Teerlink, 2020**). Addition of OM decreased the in vitro actin filament gliding rate of WT-myosin (**Figure 4A**, black points, **Table 1**, and **Video 1**), consistent with earlier measurements (**Winkelmann et al., 2015**; **Liu et al., 2015**; **Aksel et al., 2015**; **Swenson et al., 2017**) and with our previous report that OM inhibits β-cardiac myosin motility by suppressing its working stroke (**Woody et al., 2018b**). We expected that OM would further decrease the working stroke of R712-myosin. To our great surprise, however, addition of OM to R712L-myosin rescued motility in a concentration-dependent fashion (**Figure 4A**, red points, and **Video 1**). At saturating OM concentrations, the actin gliding velocity driven by R712L-myosin (1.1 ± 0.26 µm·s$^{-1}$) was near the WT-myosin velocity in the absence of OM. The half-maximal concentration for activation of motility is 30 µM, which is considerably higher than the EC$_{50}$ for inhibition of WT-myosin (0.1 µM). This result suggests that the R712L mutation affects the OM binding site, weakening the affinity. The addition of OM to R712L-myosin only moderately affects the kinetic parameters of the ATPase cycle. The exception is the value of $K_{TF}$, the half-saturation actin subunit

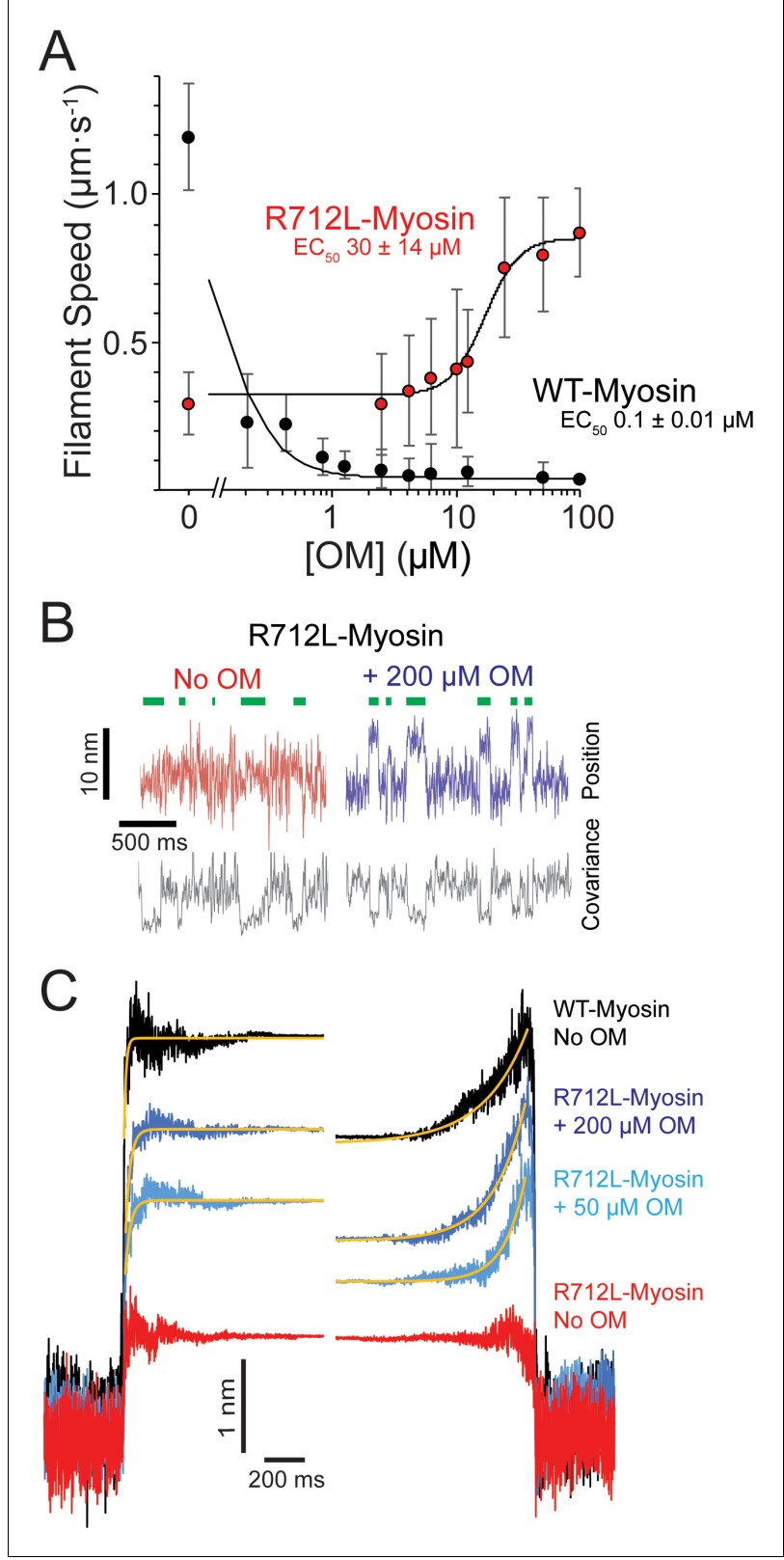

**Figure 4.** OM rescues the working stroke of R712L-myosin despite suppressing the working stroke of WT-myosin. (**A**) The speeds of individual fluorescently labeled actin filaments were quantified in gliding filament motility assays for WT-(black) and R712L-(red)-myosins as a function of omecamtiv mecarbil (OM) concentration. Speeds sharply decreased for WT-myosin with increasing OM, while a partial rescue of motility was observed for R712L-myosin at

*Figure 4 continued on next page*

*Figure 4 continued*

a higher $EC_{50}$. Errors for the $EC_{50}$ are derived from the fits of the titration data. (**B**) Single molecule displacement and covariance traces showing R712L-myosin interactions with actin dumbbells in the absence (red) and presence (blue) of 200 µM OM. Green bars indicate attachment events as detected by covariance (grey) decreases. Displacements were substantially larger upon addition of 200 µM OM. (**C**) Ensemble averages of single-molecule interactions synchronized at event beginnings and averaged forward in time (left) or synchronized at event ends and averaged backward in time (right). The ensemble averages for WT-and R712L-myosins in the absence of OM are replotted from *Figure 3* for comparison. Yellow lines are single-exponential fits to the data (*Table 2*). The online version of this article includes the following source data and figure supplement(s) for figure 4:

**Source data 1.** Excel files with data from *Figure 4*.
**Figure supplement 1.** Model for inhibition of the R712L-myosin working stroke and rescue by OM.

concentration for activation of the ATPase, which increases with OM (*Figure 1—figure supplement 3*). If phosphate release rate is limiting for the ATPase cycle, then the $K_{ATPase}$ would also be expected to increase with OM. However surprisingly, the $K_{ATPase}$ values were similar for R712L-myosin with and without OM, suggesting that the $K_{ATPase}$ and $K_{TF}$ must not be defined by the same rate constants. Indeed, previous work has suggested that the $K_{ATPase}$ is defined by a combination of rate and equilibrium constants, including phosphate release, the equilibrium constant for ATP hydrolysis, and the affinities of M.ATP and M.ADP.Pi states for actin (*White et al., 1997*).

Given the unexpected result that OM rescues gliding motility, we measured the R712L-myosin working stroke displacement and kinetics in the optical trap. Unidirectional displacements of the dumbbell upon R712L myosin–actin interaction were observed in the presence of OM. At 50 and 200 µM OM, the working stroke of R712L was increased to 2.7 and 3.4 nm, respectively (*Figure 4B, C*, light/dark blue traces, and *Table 2*). Detachment rates were independent of the OM concentration. Because the biochemical rate constants of the ATPase cycle for R712L-myosin were largely unchanged in the presence of OM (*Figure 1—figure supplements 3–8*), the concentration-dependent rescue of R712L-myosin motility can be attributed to restoration of the mechanical working stroke.

Ensemble averaging of events recorded with R712L-myosin in the presence of OM at 1 µM MgATP revealed that the working stroke is composed of two substeps like WT myosin in the absence of OM. At 50 µM OM, R712L-myosin exhibited a 1.8 nm prompt step followed by a 0.8 nm second substep (for a total working stroke of 2.7 nm). At 200 µM OM, R712L-myosin has a 2.3 nm first step followed by a 1.1 nm second substep (total working stroke: 3.4 nm) (*Figure 4C* and *Table 2*). The rising phases of time-forward averages fit a single exponential function with rates of 40–42 $s^{-1}$ at 50–200 µM OM (*Figure 4C*: left, yellow fitted curves, and *Table 2*), which is slower than that observed with WT-myosin and substantially slower than the biochemically measured rate of

**Table 3.** Exponential fits to attachment durations.
Uncertainties are 95% confidence interval limits from bootstrapping.

| | ATP (µM) | $k_1$ (s$^{-1}$) | $k_2$ (s$^{-1}$) | A |
|---|---|---|---|---|
| Wild type | 1 | 6.89 +0.7/- 0.6 | – | – |
| | 1000 | 54.4 +5.7/-4.8 | 7.5 +3.8/-2.4 | 0.97 +0.02/-0.01 |
| R712L | 1 | 7.56 +0.3/-2.4 | – | – |
| | 1000 | 69.7 +9.0/-6.7 | 7.4 +1.2/-1.0 | 0.91 +0.01/-0.01 |
| R712 + omecamtiv mecarbil (50 µM) | 1 | 7.9 +0.3/- 0.2 | – | – |
| | 1000 | 63.8 +7.8/-6.5 | 6.0 +1.9/-1.4 | 0.94 +0.08/-0.07 |

[*]Data are plotted in *Figure 3C-D*.

ADP release (157 s$^{-1}$; *Table 1*). The rising phases of time-reversed ensemble averages were adequately fit by single exponential functions (~6–10 s$^{-1}$ at 50–200 μM OM), (*Figure 4C*: right, yellow fitted curves, and *Table 2*), consistent with ATP binding rates (*Table 1*).

## Rescue of R712L by OM is reversible

To test whether rescue of the R712L working stroke by OM is reversible, we designed a flow chamber that allowed for the exchange of buffers in real time while maintaining the interrogation of single actomyosin interactions (*Figure 5A*). These chambers were prepared with highly parallel side walls facilitating very smooth flow along the direction of the actin filament with a push-pull, stepper-motor driven syringe pump. The experiment with R712L in *Figure 5B* was initiated in the presence of 50 μM OM to obtain clear displacement events (*Figure 5B*, blue trace). After acquiring an adequate number of actomyosin events, the buffer was exchanged to remove the OM, carefully maintaining the pedestal position using a camera-based stage stabilization system (*Capitanio et al., 2005*; *Woody et al., 2018a*). We found that individual myosin molecules would continue to interact with the actin dumbbell through exchange of solutions. Subsequently, the flow displaced the dumbbell slightly, in the amount expected from the Stokes drag at the fluid velocity, but the beads returned to their previous positions when the flow ended. Following this exchange, additional data were collected in the OM-free condition (*Figure 5B*, red trace). The working stroke decreased following the removal of the OM for all molecules tested (*Figure 5D*, black arrowheads). In the reverse experiment, collecting data first with an OM-free buffer, and then adding 50 μM OM by exchanging the solution, we observed rescue of the working stroke following buffer exchange (*Figure 5D*, white arrowheads). Finally, we successfully switched buffers back and forth from 50 μM OM to no OM, and partially back to the 50 μM OM solution. The molecule illustrated started with clearly discernable working strokes in the presence of OM (*Figure 5B*, blue trace), which was attenuated in the absence of OM (*Figure 5B*, red trace), and then substantially rescued following partial re-addition of OM (*Figure 5B*, purple trace; panel D, gray arrowhead). Ensemble averaging of records from each of these molecules revealed two substeps in the working stroke in each case in the presence of OM; however, the same molecules measured in the absence of OM had working strokes with single steps as evidenced by similar extension points of the time-forward and time-reversed ensemble averages (*Figure 5C*). These buffer exchange assays reveal the reversibility of working stroke rescue by OM and also very clearly show the loss of the second substep in individual myosin molecules.

## Discussion

### HCM mutation, R712L, has reduced motility due to reduced working stroke

In the present work, we analyzed a recombinant β-cardiac myosin that contains a highly penetrant R712L mutation that causes severe HCM (*Sakthivel et al., 2000*). R712 is located in the converter region at the fulcrum of the lever arm in the myosin head and forms a salt bridge with relay helix E497, likely stabilizing the lever at a functionally crucial region of the motor and maintaining mechanical integrity when the converter and lever arm tilt (*Figures 1A* and *2B*). Given the importance of this region in the mechanochemistry of myosin, we investigated the effect of the mutation on key steps of the actomyosin ATPase pathway, in vitro actin gliding, and the working stroke displacement and kinetics. R712L-myosin has a drastically reduced actin gliding rate and a markedly reduced mechanical working stroke, despite minimal alteration of rates of the biochemical steps in its actomyosin ATPase cycle. The suppressed working stroke explains the reduced filament gliding velocity and presumably the suppressed cardiac performance in the disease. MD simulations support the concept that disruption of the R712-E497 salt bridge destabilizes the interaction between the converter domain and the relay helix, thereby decoupling ATPase activity from work output.

### The working stroke of R712L-myosin is defective

Ensemble averages of single R712L-myosin interactions show a substantially smaller average displacement (1.3 nm) compared to WT-myosin (4.4 nm; *Figure 2B* and *Table 2*). Importantly, averages of R712L-myosin do not reveal a two-step working stroke, but rather show a single, small displacement immediately upon actin attachment (*Figure 4C* and *5C*—, red traces). Events synchronized at

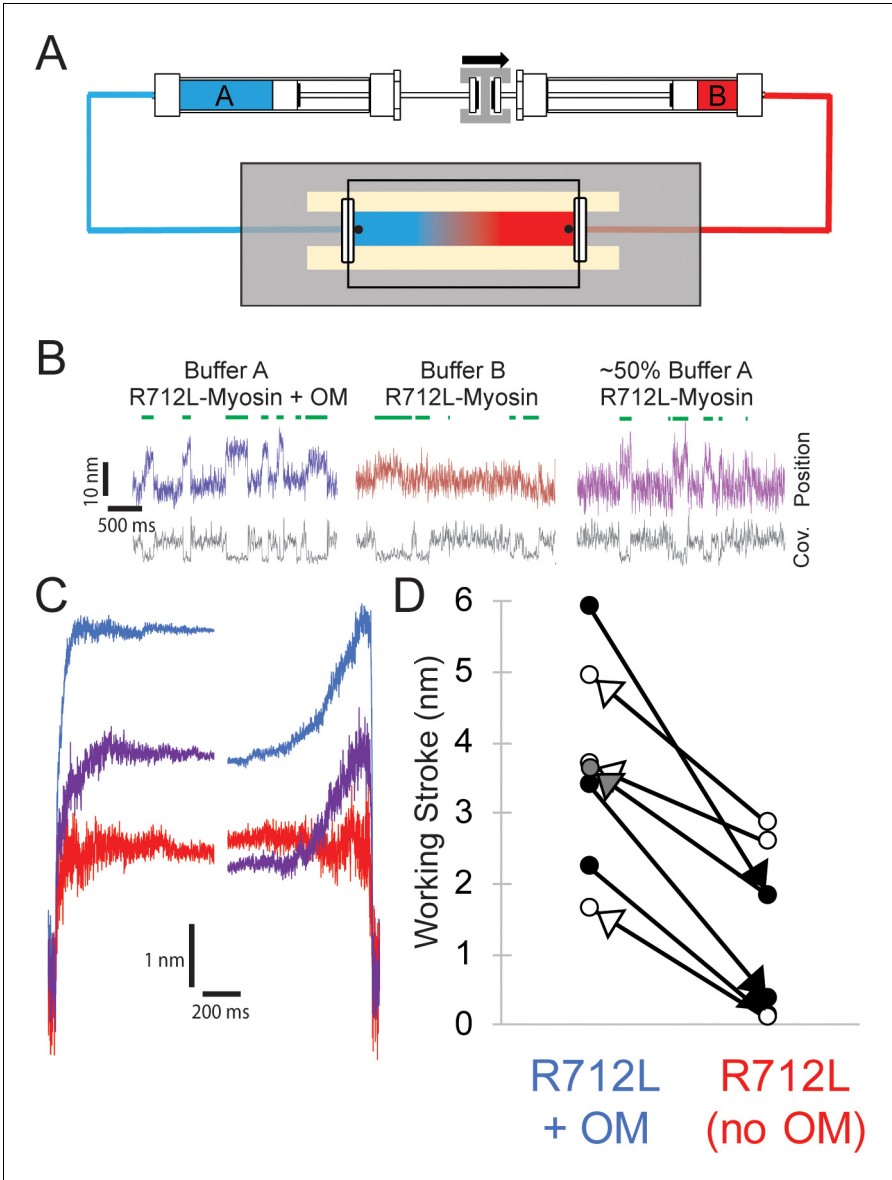

**Figure 5.** Reversible rescue of a single R712L-myosin molecule by OM as revealed by real-time buffer exchange. (A) Diagram of flow cell and back-to-back syringes used for buffer exchange experiments during optical trapping assays (see Materials and methods). A single molecule is initially interrogated with 'buffer A' (blue) in the chamber, which is exchanged with 'buffer B' (red) at a rate of ~0.5 chamber volumes per minute via action of a push-pull syringe pump (an average linear flow rate of ~0.5 mm·s$^{-1}$). Precise control of pushing and extracting the buffer, in addition to maintenance of the stage with positional feedback, allows analysis of the same molecule under different solution conditions. (B) Example traces of an individual myosin molecule under multiple buffer conditions. (Left, blue) The initial trace showing displacements in the presence of OM, followed by attachment events with smaller displacements in OM-free buffer (center, red), followed again by a partial rescue of working strokes as the OM-containing buffer reversed and re-entered the chamber (right, purple). (C) Ensemble averages of the molecule from (B), with a shortened working stroke upon washout of OM, and a partial rescue of the working stroke after partial re-addition of OM. (D) Working stroke amplitude in the presence and absence of OM from flow experiments. Each line represents the change in total observed working stroke via ensemble averaging of records from an individual molecule under multiple buffer conditions. Black arrowheads indicate experiments where buffer A contained OM and buffer B lacked OM, and in each case a reduced working stroke was observed. White arrowheads indicate experiments where buffer A lacked OM and buffer B contained OM. In each case addition of OM increased the working stroke. The gray arrowhead depicts the partial rescue experiment in which a mixture of Buffer A and buffer B was in the chamber after reversal of the flow.

*Figure 5 continued on next page*

*Figure 5 continued*
The online version of this article includes the following source data for figure 5:
**Source data 1.** Excel files with data from *Figure 5*.

the end of attachment (reverse-ensemble averages) showed no kinetic rise before detachment, also supporting the absence of a second substep (*Figure 4C*). In WT-myosin, an initial displacement (~3–4 nm) occurs upon strong actin binding and precedes release of orthophosphate ($P_i$) from actin-myosin-ADP-$P_i$ (*Woody et al., 2019*). This step is followed by a 1–2 nm displacement that takes place upon release of ADP from actin-myosin-ADP, which confers force dependence of actin detachment (*Woody et al., 2019*; *Veigel et al., 2003*; *Greenberg et al., 2014*). Because the R712L-myosin produces its small working stroke immediately upon attachment, and the $P_i$ release rate measured in biochemical experiments is not inhibited (*Table 1*), we propose that the small working stroke in the mutant is linked to $P_i$ release rather than to ADP release. However, the smaller displacement and the absence of a second step indicate that the normal coupling between the active site and lever arm is substantially diminished. This loss of the second step may result in altered mechanosensitivity of myosin. Further characterization of R712L-myosin under load will be important to fully understand the mechanism of action of this mutation.

An alternative hypothesis for the observed smaller average displacement is that myosin alternates between a state having a normal displacement and a state with a defective working stroke. However, we disfavor this possibility given that the ensemble averages reveal a one-step working stroke, rather than the sum of two reduced displacements (*Figure 3D* and *Table 3*).

## R712L working stroke is rescued by OM despite little effect on ATPase activity

A remarkable finding of this study is that OM rescues the working stroke and actin gliding activity of R712L-myosin without substantially changing myosin's biochemical kinetics (*Figures 4B,C* and *5B–D*, and *Table 1*). OM was identified in a high-throughput screen designed to identify drugs that increase the rate of $P_I$ release from actomyosin-ADP-$P_i$, with the goal of increasing the β-cardiac myosin duty ratio and heart contractility (*Malik et al., 2011*). It was subsequently found that OM increases cardiac contractility by indirect activation of the muscle thin filament (*Woody et al., 2018b*; *Governali et al., 2020*). OM suppresses the working stroke of WT-myosin, similar to the effect of the R712L mutation. OM prolongs the actomyosin attachment time, which leads to cooperative sensitization of the TF regulatory system to $Ca^{2+}$, thereby enhancing contraction (*Woody et al., 2018b*; *Liu et al., 2018*). We termed this combination of effects SEPTA (step eliminated, prolonged time of attachment; *Woody et al., 2018b*).

We expected OM to have similar effects on R712L-myosin as it did in WT-myosin: to increase the rate of $P_i$ release from R712L-myosin and to further suppress the already defective actin gliding velocity and working stroke. However, we found *rescue* of actin gliding in the in vitro motility assay (*Figure 4A*) due to the near-complete recovery of the R712L-myosin working stroke (*Figure 4B,C*) with little change to the ATPase activity (*Table 1*).

Ten-fold higher OM concentrations were required to achieve rescue of R712L-myosin than to inhibit WT-myosin, suggesting the R712L mutation alters OM affinity. Structural studies with WT-myosin found two different OM binding conformations that depend on whether the motor is in a pre-powerstroke or a post-powerstroke state, with a 10-fold tighter affinity for the pre-powerstroke state (*Winkelmann et al., 2015*; *Planelles-Herrero et al., 2017*). R712 is a key component of the OM binding site in both conformations, with OM forming packing interactions and shielding the R712-E497 salt bridge in the lever arm converter region.

Biochemical kinetics experiments show that OM does not increase actin-activated phosphate release from R712L-myosin as found for WT-myosin (*Table 1*). This result suggests that OM may not bind to the pre-powerstroke state of R712L-myosin, which is the higher affinity state in WT-myosin. Thus, the higher $EC_{50}$ for R712L-myosin may be the result of a disrupted OM binding site, thereby causing OM to bind only to the lower affinity post-powerstroke state. A surprising and counterintuitive finding of this study is that OM binding does not suppress the R712L working stroke, and it

does not result in a prolonged time of attachment (SEPTA; *Woody et al., 2018b*). We suggest that SEPTA is the result of binding of OM to the pre-powerstroke state of WT-myosin.

How does OM rescue the R712L-myosin working stroke? We propose that OM binds to post-powerstroke state of R712L-myosin at the same site as WT-myosin, and its presence restores the mechanical integrity of this junction, reconnecting biochemical and mechanical activity. Binding of OM to R712L-myosin not only increases the size of the initial displacement that occurs upon actin binding, it also rescues the second substep (*Figure 4—figure supplement 1*). MD simulations suggest that the R712L mutation may disrupt the OM binding site (*Video 4*) and alter the rigid coupling between the converter/lever arm and motor. This mechanical disruption is prevented in the presence of OM, stabilizing the WT-like configuration needed for displacement and force generation.

## Rescue of R712L working stroke by OM is reversible

Tests of the reversibility of the effects of OM on R712L-myosin were initially hampered by the large number of recordings necessary to obtain statistically reliable effects of adding and removing the drug in separate molecules. Intrinsic variability among optical trap recordings of working stroke displacements and kinetics is inevitable due to the probabilistic nature of the mechanical strain at time of attachment, caused by Brownian fluctuations of the bead-actin-bead dumbbell and, possibly, due to protein heterogeneity (*Finer et al., 1994*; *Steffen et al., 2001*). This problem prompted us to design a flow chamber that would enable exchange of buffers while continuing to analyze single actomyosin dumbbells. This assay allowed us to unambiguously demonstrate that the defective working stroke of a single R712L myosin can be rescued by OM binding, and that this rescue is reversible. Notably, at the intermediate OM concentration examined (<50 μM), the average displacement of a single R712L-myosin molecule was in between the values obtained in the absence and presence of 50 μM OM, which suggests that OM was binding and dissociating during the acquisition of the trace.

Exchange of solutions is commonly applied in other single-molecule experiments, such as with DNA-binding proteins, where the sample can be moved into different flow streams (*Gross et al., 2010*; *Forget et al., 2013*). This method is not applicable to the actomyosin three-bead assay, however, because the pedestal bead containing the myosin is attached to the microscope slide. To our knowledge, this is the first successful exchange of solutions while maintaining interrogation of an individual actomyosin pair with the three-bead assay. Further microfluidic improvements to the flow chamber should provide for more rapid comparison of conditions with increased statistical power.

## The R712L mutation and hypertrophic cardiomyopathy

The R712L mutation very clearly results in suppression of motor function by uncoupling the ATPase activity of the myosin from its working stroke. We would expect that the contractility powered by a thick filament that contains R712L-myosins would exhibit decreased sliding velocity, force, and power output. Thus, simply considering the myosin activity, the R712L mutation does not fit into the paradigm expressed for several other mutations in which HCM mutations result in gain-of-function contractility.

Although a mechanism is not apparent, it is possible that disruption of force dependence in myosin may be a factor in the HCM phenotype, but further studies will be required to analyze this possibility. It is possible that in the context of the sarcomere of a heterozygous individual, the R712L mutation could result in a gain of function. As discussed above, we recently discovered that although OM inhibits the myosin power stroke and kinetics, at therapeutic concentrations it may act as a thin-filament sensitizer, allowing increased overall contractility at lower calcium concentrations (*Woody et al., 2018b*; *Governali et al., 2020*). Likewise, R712L-myosin could conceivably confer activating properties through the TF, or perhaps through the activation of other myosin heads from the thick filament, such as through alteration of the interacting heads motif (*Spudich, 2015*). Another likely possibility is that the gain-of-function concept is not universal in HCM.

## Conclusions

We found that mutation of R712 to leucine leads to a defective working stroke in β-cardiac myosin, perhaps leading to defective cardiac contraction in this variant of HCM. OM rescues the defective

working stroke of the mutant, and this surprising effect of OM is reversible upon exchanging buffers in individual myosin-actin dumbbell interaction sites.

## Materials and methods

### Protein expression and purification

Adenovirus manipulation

The human β-cardiac HMM (cHMM) encodes residues 1–1137 of the *MYH7* gene (GenBank: AAA51837.1) with a FLAG tag added on the C-terminus (1138–1146) of the S2 domain. The cHMM cDNA was cloned into the pShuttle-IRES-hrGFP-1 vector (Agilent Tech., Santa Clara, CA) and an AdcHMM-Flag virus was prepared and amplified for expression of cHMM protein in C2C12 cells (*Luo et al., 2007*). The virus was expanded by infection of a large number of plates of confluent Ad293 cells at multiplicity of infection (MOI) of 3–5. The virus was harvested from the cells and purified by CsCl density sedimentation yielding final virus titers of $10^{10}$–$10^{11}$ plaque forming units per mL (pfu·mL$^{-1}$).

### Muscle cell expression and purification of β-cardiac HMM

Maintenance of the mouse myogenic cell line, C2C12 (CRL 1772; American Type Culture Collection, Rockville, MD), has been described in detail elsewhere (*Chow et al., 2002*; *Wang et al., 2003*). Confluent C2C12 myoblasts were infected with replication defective recombinant adenovirus (AdcHMM-Flag) at $2.7 \times 10^8$ pfu·mL$^{-1}$ in fusion medium (89% DMEM, 10% horse serum, 1% FBS). Expression of recombinant cHMM was monitored by accumulation of co-expressed GFP fluorescence in infected cells. Myocyte differentiation and GFP accumulation were monitored for 216–264 hr after which the cells were harvested. Cells were chilled, media removed, and the cell layer was rinsed with cold PBS. The cell layer was scraped into Triton extraction buffer: 100 mM NaCl, 0.5% Triton X-100, 10 mM Imidazole pH 7.0, 1 mM DTT, 5 mM MgATP, and protease inhibitor cocktail (Sigma, St. Louis, MO). The cell suspension was collected in an ice-cold Dounce homogenizer and lysed with 15 strokes of the tight pestle. The cell debris in the whole cell lysate was pelleted by centrifugation at 17,000 x *g* for 15 min at 4°C. The Triton soluble extract was fractionated by ammonium sulfate precipitation using sequential steps of 0–30% saturation and 30–60% saturation. The cHMM precipitates between 30–60% saturation of ammonium sulfate. The recovered pellet was dissolved in and dialyzed against 10 mM Imidazole, 150 mM NaCl, pH 7.4 for affinity purification of the FLAG-tagged cHMM on M2 mAb-Sepharose beads (Sigma). Bound cHMM was eluted with 0.1 mg·mL$^{-1}$ FLAG peptide (Sigma). Protein was concentrated and buffer exchanged on Amicon Ultracel-10K centrifugal filters (Millipore; Darmstadt, Germany), dialyzed exhaustively into 10 mM MOPS, 100 mM KCl, 1 mM DTT before a final centrifugation at 300,000 x *g* for 10 min at 4°C. Aliquots were drop frozen in liquid nitrogen and stored in vapor phase at –147°C.

### SDS-PAGE and LC/MS-MS sequence analysis of the expressed cHMM

Purified WT human β-cHMM and R712L HCM variants were routinely analyzed by SDS-PAGE (*Figure 1—figure supplement 1A–B*). The purified proteins, which we call WT-myosin and R712L-myosin, consist of a 132 kDa heavy chain and associated myosin light chains LC1 and LC2. The protein sequence of the expressed WT- and R712L-myosins were determined by LC/MS-MS analysis of independent trypsin and chymotrypsin digests of WT and mutated proteins. The peptide coverage was complete and comparable for each protein from the N-terminal acetylated glycine through the C-terminal Flag-tag (2–1146) and differed only in the unique peptides that confirm the single residue substitutions distinguishing the WT and the mutated proteins.

### Reagents

Actin was purified from rabbit skeletal muscle (*Spudich and Watt, 1971*). Native porcine cardiac TFs were prepared according to the procedure of *Spiess et al., 1999* as modified by *Matsumoto et al., 2004*. ATP and ADP were purchased from Sigma-Aldrich. OM (CK-1827452) was purchased from Selleck Chemicals. A 20 mM stock solution of OM was prepared in dimethyl sulfoxide (DMSO) and aliquots were stored at −80°C. *N*-[2-(1-maleimidyl)ethyl]−7-(diethylamino)coumarin-3-carboxamide-labeled phosphate binding protein (MDCC-PBP) was prepared according to *Brune et al., 1994*.

## Motility assays

Measurement of in vitro gliding filament motility of human β-cardiac HMM was done as previously described (*Winkelmann et al., 2015*). Briefly, β-cHMM was bound to nitrocellulose coated coverslips for 2 min, loading at 10–100 µg·mL$^{-1}$. The surfaces were blocked with 1% bovine serum albumin (BSA) in PBS for 5 min. Motility was measured in a 12 µl assay chamber in motility buffer (25 mM Imidazole, 25 mM KCl, 4 mM MgCl$_2$, 7.5 mM MgATP, 0.5% methyl cellulose, 0.1 mg·mL$^{-1}$ glucose oxidase, 0.018 mg·mL$^{-1}$ catalase, 2.3 mg·mL$^{-1}$ glucose, and 5 mM DTT, pH 7.6) containing 1 nM phalloidin–rhodamine-labeled actin (rhodamine-phalloidin; Sigma). To titrate the effect of the drug on motility, a 2.5 mM stock of OM (CK-1827452) in dimethylsulphoxide (DMSO) was serially diluted with DMSO before a final 1/200 dilution into motility buffer containing the rhodamine-labeled actin. The 0.5% DMSO in the assay buffer had no effect on motility in the absence of drug. The chamber was observed with a temperature-controlled stage and objective set at 32°C on an upright microscope with an image-intensified charge-coupled device camera capturing data to an acquisition computer at 5–30 fps. dependent on assay parameters. Movement of actin filaments from 500 to 1000 frames of continuous imaging was analyzed with semi-automated filament tracking programs as previously described (*Barua et al., 2012*; *Bourdieu et al., 1995*). The trajectory of every filament with a lifetime of at least 10 frames was determined; the instantaneous velocity of the filament moving along the trajectory, the filament length, the distance of continuous motion, and the duration of pauses were tabulated. A weighted probability of the actin filament velocity for hundreds of events was fit by a Gaussian distribution and reported as a mean velocity and standard deviation for each experimental condition.

## Biochemical characterization

### TF-activated steady-state ATPase measurements

Steady-state ATPase activity was measured by an NADH coupled assay as described previously (*Haithcock et al., 2011*). Addition of DMSO (0.25–2%) had no effect on the rates. Measurements with TFs were carried out at pCa <4 (100 µM Ca) and the KCl concentration was kept at <0.05 mM. The ATPase activity with TFs alone was subtracted from the data obtained in the experiments which were done with myosin plus TFs.

### Stopped-flow experiments

All stopped-flow measurements were performed at 20°C using a Hi-tech Scientific SF-61S×2 stopped-flow system equipped with a 75 W mercury-xenon arc lamp. Single mixing experiments resulted in a 1:2 dilution of myosin or actomyosin +/− ADP and a 1:2 dilution of nucleotide in the flow cell in a buffer containing 5 mM MOPS (pH 7.2), 2 mM MgCl$_2$, 25 mM KCl. In double mixing experiments myosin and ATP were mixed, allowed to incubate for the desired time, and then mixed with TFs to give a 1:4 dilution of myosin and nucleotide and a 1:2 dilution of TFs in the flow cell in a buffer containing 5 mM MOPS (pH 7.2), 2 mM MgCl$_2$, 10 mM KCl. All syringes contained either 0.25% DMSO or 50 µM OM. Light scattering was measured using an excitation wavelength of 432 nm and a 400 nm longpass filter. Tryptophan fluorescence experiments utilized excitation at 295 nm and emission was selected with a 320–380 nm bandpass filter. Phosphate dissociation from the TF myosin ADP-P$_i$ complex was measured using MDCC-PBP as described by *White et al., 1997* with excitation wavelength of 434 nm and a 455 nm long-pass filter. Background P$_i$ was removed by including a phosphate mop consisting of 0.10 mM 7-methylguanosine and 0.02 units·mL$^{-1}$ purine-nucleoside phosphorylase (Sigma) in all of the reaction solutions. TF and myosin solutions were extensively dialyzed (>3 times against a 1000-fold volume of buffer). The pH of buffers used in phosphate dissociation experiments was adjusted by adding 1 N sodium hydroxide after which a small sample of buffer was used to determine the pH, then discarded to avoid contaminating the buffer with phosphate from the pH.

### Data analysis and kinetic simulation of stopped-flow data

Three to four data sets of 1024 point recordings were averaged, and the observed rate constants were obtained by fitting one or two exponential equations to the data using the TgK Scientific Kinetic Studio 5.10 software package included with the Hi-tech stopped-flow instrument (Bradford-on Avon, UK).

## Flow cells and optical trapping

### Flow cell chambers

We constructed flow cell chambers with double-sided tape and vacuum grease as previously described (*Greenberg et al., 2017*). Briefly, the surface of the coverslip was coated with a 0.1% nitrocellulose solution (Electron Microscopy Sciences) mixed with 2.47-µm-diameter silica beads. Nitrocellulose was allowed to dry on the coverslip for at least 30 min, and the coverslips were used within 24 hr of preparation or were stored in vacuum-sealed bags at 4°C until further use. To define two walls of the flow cell, two strips of double-sided tape were placed 0.5 cm apart onto the glass coverslip, and then a 1-mm-thick glass slide was placed onto the tape and carefully sealed.

Trapping buffer (25 mM KCl, 60 mM MOPS, 1 mM DTT, 1 mM MgCl$_2$, 1 mM EGTA) was used as the solvent for all components in the optical trapping assay, unless otherwise noted. β-cardiac myosin variants were stored and diluted in trapping buffer with 300 mM added KCl. Cardiac myosin was added to the chamber and allowed to nonspecifically adsorb to the nitrocellulose surface for 30 s. The loading concentration of β-cardiac myosin ranged between 0.02 and 0.1 µg·mL$^{-1}$ and was adjusted daily such that 1 of 3–5 locations tested showed clear interactions with the actin dumbbell. Immediately following the 30 s myosin incubation, chambers were blocked with two, 3 min incubations of 1 mg·mL$^{-1}$BSA. Following blocking, trapping buffer was added to the chamber with indicated amounts of MgATP, 50–200 µM OM in DMSO (or 0.5–2% DMSO for control experiments), and 0.1–0.25 nM rabbit skeletal actin filaments polymerized with 10–15% biotinylated actin (Cytoskeleton) and stabilized by rhodamine-phalloidin (Sigma) at 1.1–1.2 molar ratio with actin monomers. Then, 100x stocks of glucose oxidase + catalase (GOC) were freshly prepared by centrifuging catalase from bovine liver (Sigma) at 15,000 x *g* for 1 min, and adding 2 µl of catalase supernatant to 20 µl of 19.1 U·µL$^{-1}$ of glucose oxidase (Sigma). Immediately prior to addition of trapping buffer to the chamber, 1 µL of 250 mg·mL$^{-1}$ of glucose and 1 µL of 100x GOC were added to 98 µl of trapping buffer (for final amount of 1x GOC solution and 2.5 mg·mL$^{-1}$ glucose) (*Greenberg et al., 2017*). Low ATP concentrations were verified by absorbance at 259 nm (extinction coefficient 15.4 × 10$^{-3}$ M$^{-1}$cm$^{-1}$); 0.4 ng of 500-nm-diameter polystyrene beads (Polysciences) was coated with 5 mg·mL$^{-1}$ neutravidin solution (Thermo Fisher) overnight at 4°C; 3 µL of coated beads was added to one side of the chamber. After addition of the assay components, the flow cell was sealed with vacuum grease. For flow cells used in the buffer exchange experiments, a thin layer of UV-curable resin (Loon) was brushed onto the top of the vacuum grease and cured for 5–10 s with an ultraviolet lamp to reduce leakiness of the chambers under flow.

## Optical trapping assay

Optical trapping experiments were performed as previously described (*Woody et al., 2018b*) in a dual-beam optical trap with a 1064 nm trapping laser. A Nikon Plan Apo x60 water immersion objective (NA 1.2) and Nikon HNA oil condenser lens were used in the microscope. Force detection was measured directly with quadrant photodiode detectors (JQ-50P, Electro Optical Components Inc, Santa Rosa, CA) with high-voltage reverse bias with an amplifier custom built for our setup (*Woody et al., 2018a*). Two beams were produced by a polarizing beam splitter. The 500 nm beads were trapped, one in each beam, approximately 5 µm apart with trap stiffness of 0.05–0.07 pN·nm$^{-1}$ (as calculated via the power spectrum of each bead). Next, a fluorescently labeled actin filament of approximate length ~5–10 µm was tethered between the two beads. A pretension of ~3–5 pN was applied to the actin filament, and this bead-actin-bead 'dumbbell' was used to search for the presence of β-cardiac myosin on pedestal beads. Interactions with β-cardiac myosin could be detected by both a decrease in covariance of the two bead positions and power stroke deflection of the beads within the trap. Once a putative molecule was identified, the dumbbell was carefully positioned over the molecule such that it produced maximal deflections and interacted with the greatest frequency, and this position was maintained by a feedback system that stabilized the position of the stage based on images of the pedestal beads. We used custom-built programs (Labview, Matlab) to acquire data and calculate the feedback signal at 250 kHz. During acquisitions, we manually adjusted the position of the stage in steps of 6 nm axially along the dumbbell between acquisition traces to ensure even accessibility of actin-attachment target zones.

## Optical trap data analysis

We analyzed the optical trap data from force signals of the two beads as previously described (*Woody et al., 2018b*; *Greenberg et al., 2017*; *Chen et al., 2012*). Briefly, we detected events by calculating the covariance of the two beads' positions using an averaging window of 20–30 ms. The distribution of covariances from a 15 s recording of myosin interactions was well described by two Gaussian distributions. The first Gaussian peak at the lower covariance value is associated with the bound state of the actin dumbbell to the myosin molecule, while the second peak at higher covariance represents the covariance of the dumbbell beads while myosin was detached. The minimum detectable event time for each molecule studied (the dead time) was determined to be half of the covariance averaging window. This window was adjusted to be as low as possible while maintaining separation between bound and unbound peaks such that the unbound peak mean minus its standard deviation was greater than the bound peak plus its standard deviation. Molecules where this separation could not be achieved were not analyzed further.

## Actomyosin binding events were identified and refined in a two-step process

First, in order to minimize false positives, events were selected where the covariance crossed from the average unbound covariance peak to the average of the bound covariance peak and back again. The start and end times were initially recorded where the covariance signal first crossed the average bound and unbound values, respectively. Next, as the covariance signal is a slightly delayed indicator of attachment and detachment, the event start was further refined by determining when the covariance trace first crossed below the value halfway between the bound and unbound peaks near the initial beginning marker of each event. 'Near' the originally detected event was defined as within 1.5x the instrument dead time or within the duration of the detected event, whichever was smaller. Event ends were refined similarly to the event beginnings, that the refined event ends were marked to the first point in time at which the covariance trace crossed above 80% of the way back toward the unbound peak. For display and further calculation, event starts and ends were shifted minus or plus 0.75x the dead time, respectively, to account for the effects of calculating the covariance using the finite averaging window.

Event durations were defined as the interval between these refined start and end times. Events shorter than the calculated instrument dead time were excluded from analysis. Ensemble averages were performed by aligning events at their refined beginnings (time-forward ensemble averages) or at their refined ends (time-reversed ensemble averages). To facilitate the averaging of events with various lengths, the displacement values for each of the interactions were extended forward or backward in time using the displacement of the interaction immediately before the event end or after the event start for time-forward and time-reversed ensemble averages, respectively. The total working stroke size was determined by subtracting the minimum position of the beads immediately prior to attachment from the time-forward extensions. The minimum value of the trace was determined with an 8 ms averaging window within +/− 0.2 s of the detected event start. The second substep of actomyosin displacement was determined by subtracting the total step size from the averaged extensions of the time-reversed ensemble averages. Signals were weighted such that each molecule contributed equally to averages.

## Attachment duration and step size parameter estimation

As previously described, we used MEMLET to estimate detachment rates and mean step sizes (*Woody et al., 2016*), which allowed us to perform maximum-likelihood estimation without the need for binning. Only molecules which had >75 events were included in analysis.

## Buffer exchange experiments

Buffer exchange optical trapping experiments were performed as above, except custom flow chambers were used instead of fully sealed chambers. Holes were drilled in the 1 mm glass slides using a diamond-tipped drill bit, and PEEK tubing was inserted into the holes and sealed with UV-cured resin (Loon). Following insertion of the PEEK tubing, we used a razor blade to trim the tubing carefully to ensure the tubing was flush with the inner plane of the glass slide. Buffer A was used in the chamber, and a second buffer, 'buffer B', was prepared in a syringe and attached via PEEK tubing to

the flow cell. Buffer B was the same as buffer A, but contained a different dependent (i.e., if buffer A contained OM, buffer B contained background DMSO, and vice versa). Volumes were carefully measured for each flow cell by addition of buffer A prior to sealing the chamber, and these volumes were used to adjust the flow rate of the syringe pump. The entrance syringe was loaded onto the 'infuse' side of a continuous flow push-pull syringe pump (KD Scientific 260 Legacy, Holliston, MA), and an oppositely oriented exit syringe was loaded onto the 'withdraw' side of the syringe pump. Data were first acquired with buffer A in the chamber, then the syringe pump was turned on to create a flow of ~0.5 chamber volumes per minute, which resulted in an average flow rate of ~0.5 mm·s$^{-1}$. Due to the simultaneous motion of 'infuse' and 'withdraw' syringes, the buffer was gently exchanged with no change in pressure or volume. To fully exchange buffers, two full chamber volumes were flowed. After exchange, the pump was switched off, and acquisitions were restarted with the dumbbell in the same position over the pedestal bead. In the experiment where we partially reversed flow (*Figure 5B,C*), the flow of the syringe pump was reversed flowing back 1.5 chamber volumes so that the chamber contained a mixture of buffer A and buffer B. Then data were acquired on the same molecule.

## MD simulations setup and analysis

The starting conformation for MD simulations was the crystal structure of the post-rigor conformation of β-cardiac myosin, PDB file 6FSA (*Robert-Paganin et al., 2018*). MD and steered MD simulations were performed with the GPU-based NAMD package. The CHARMM36 parameter set was used for the protein and TIP3P model for the water molecules (*Jorgensen et al., 1983*). With all other atoms fixed, the waters were energy minimized for 20 ps and equilibrated with NVT (constant number [N], volume [V], and temperature [T]) run for 1.0 ns at 320 K. The full model was then minimized for 20 ps in one fs steps and then equilibrated at constant NPT (constant number [N], pressure [P], and temperature [T]), 1.0 ATM (1.01325 bar), and 320 K for 100 ns without constraints.

Steered MD simulations were performed at constant NPT, 1.0 ATM and 320 K for 125 ns with the actin binding loops of the β-cardiac myosin (residue numbers: 363–376, 401–415, 540–544, 557–577, and 623–647) fixed and steering force applied to the Cα atom of heavy chain residue 788 at the center of the essential light chain binding domain. The direction of the applied force was along the direction of a hypothetically bound actin filament as determined by aligning the cardiac myosin of the last frame of the equilibration with a high resolution cryoEM structure of rigor myosin 1b bound to actin, 6C1H (*Mentes et al., 2018*). The system involves a total of ~187 k atoms in a 9.5 × 8.4×16.2 nm$^3$ solvent box with free K$^+$ and Cl$^-$ concentrations of 150 mM.

Angular positions of the myosin lever arm were determined using the colvars module (*Fiorin et al., 2013*) of VMD (*Humphrey et al., 1996*) and colvar functions (components) *tilt*, for the axial angle and *spinAngle* for azimuthal rotation. The inclination is the angle between the actin filament axis (the Z-axis shown as blue in the video) and a vector along the lever arm helix (LAH) between residues 768–788. The plot shows the unloaded time-zero inclination angle (127°) minus the loaded inclination angle. The azimuth is the angle between the X axis and the projection of the LAH vector onto the X-Y plane (blue and green arrows). Positive azimuth angles are counter-clockwise when viewed from the pointed end toward the barbed end of the filament.

## Acknowledgements

The work was supported by NIH grants R01-HL133863 to EF, DAW, and EMO, R35GM118139 to YEG, S10OD023592 to Dr. Kim Sharp (University of Pennsylvania), and NSF grant CMMI:15-48571 to YEG and EMO. We thank Drs. Michael Woody and Serapion Pyrpassopoulos for help with the optical trap and for useful discussions.

## Additional information

### Funding

| Funder | Grant reference number | Author |
| --- | --- | --- |
| National Institutes of Health | R01-HL133863 | Eva Forgacs<br>Donald Winkelmann |

| | | E Michael Ostap |
|---|---|---|
| National Institutes of Health | R35GM118139 | Yale E Goldman |
| National Science Foundation | CMMI:15–48571 | Yale E Goldman<br>E Michael Ostap |

The funders had no role in study design, data collection and interpretation, or the decision to submit the work for publication.

## Author contributions

Aaron Snoberger, Conceptualization, Data curation, Software, Formal analysis, Validation, Investigation, Visualization, Methodology, Writing - original draft, Writing - review and editing; Bipasha Barua, Conceptualization, Resources, Data curation, Software, Formal analysis, Validation, Investigation, Visualization, Methodology, Writing - original draft, Writing - review and editing; Jennifer L Atherton, Resources, Formal analysis, Investigation, Visualization, Writing - review and editing; Henry Shuman, Data curation, Software, Formal analysis, Investigation, Visualization, Writing - review and editing; Eva Forgacs, Data curation, Software, Formal analysis, Supervision, Funding acquisition, Visualization, Writing - review and editing; Yale E Goldman, Donald A Winkelmann, E Michael Ostap, Conceptualization, Resources, Data curation, Software, Supervision, Funding acquisition, Validation, Visualization, Methodology, Writing - original draft, Project administration, Writing - review and editing

## Author ORCIDs

Yale E Goldman (iD) http://orcid.org/0000-0002-2492-9194
E Michael Ostap (iD) https://orcid.org/0000-0003-0544-9360

## Decision letter and Author response

Decision letter https://doi.org/10.7554/eLife.63691.sa1
Author response https://doi.org/10.7554/eLife.63691.sa2

## Additional files

### Supplementary files

- Transparent reporting form

### Data availability

Source data has been provided for Figures 1-5. Raw optical trapping data can be downloaded from https://zenodo.org/record/4437109 (https://doi.org/10.5281/zenodo.4437108).

The following dataset was generated:

| Author(s) | Year | Dataset title | Dataset URL | Database and Identifier |
|---|---|---|---|---|
| Snoberger A | 2021 | Optical Trapping Data for the manuscript "Myosin with hypertrophic cardiomyopathy mutation R712L has a reduced working stroke which is rescued by omecamtiv mecarbil" | https://zenodo.org/record/4437109 | Zenodo, 10.5281/zenodo.4437109 |

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
