## [Decision Letter]

**Acceptance summary:**

The data will be of interest to a wide range of scientists, including basic scientists in the cardiac and muscle fields, as well as translational scientists who seek therapeutic advances.

**Decision letter after peer review:**

Thank you for submitting your article "Hypertrophic cardiomyopathy myosin-mutation R712L suppresses myosin's power stroke and is rescued by omecamtiv mecarbil" for consideration by *eLife*. Your article has been reviewed by three peer reviewers, including Ahmet Yildiz as the Reviewing Editor and Reviewer #1, and the evaluation has been overseen by Vivek Malhotra as the Senior Editor. The following individuals involved in review of your submission have agreed to reveal their identity: David D Thomas (Reviewer #2); Michael A Geeves (Reviewer #3).

The reviewers have discussed the reviews with one another and the Reviewing Editor has drafted this decision to help you prepare a revised submission.

Summary:

This manuscript investigates the mechanochemical impact of one of the cardiomyopathy mutations on human β-cardiac myosin II. Mutations in β-cardiac myosin (MYH7) causes several cardiac diseases, such as hypertrophic cardiomyopathy (HCM) and dilated cardiomyopathy (DCM). It has been proposed that HCM arises from mutations that enhance myosin activity, whereas DCM arises from loss-of-function mutants. Consistent with this view, omecamtiv mecarbil (OM), a drug in clinical phase III trials for the treatment of heart failure, was developed as a cardiac myosin activator. However, these authors and others in the field have recently shown that OM inhibits the velocity and force generation of native myosin, which disagrees with the predictions of the originally proposed "gain of function" model of HCM and suggests a more complicated mechanism for the molecular causes of these diseases.

The manuscript focuses on the R712L mutation, which is linked to severe HCM. R712 lies at the interface of the converter/lever arm and the relay helix and can form a salt bridge with E497. In many cases, HCM is classified as a hyper contractile phenotype but the report here demonstrates that this mutation is very disruptive of normal myosin function. Many mutations cause a mild dysfunction of myosin activity but are compensated by a disruption in the regulation of contraction which results in hyper contractile activity. Understanding how this severe mutation causes HCM could be very important in understanding the pathway from mutation to disease, which can take many years to develop.

Consistent with the location of the mutation, at the site of amplification of relay helix movement to the large swing of the lever arm, there is little disruption of the ATPase cycle but a loss of coupling between the ATPase and the mechanical events. This is shown by a loss of motility and step size while the ATPase remains normal. The mutation is also close to the site of OM binding. OM is a potential heart failure drug. Unexpectedly OM recovered some of the activity (motility and step size) when bound to the mutant myosin presumably by stabilizing the pre-power stroke structure allowing a more complete working stroke. Based on these results, the authors exclude the gain of function model for OM and propose a possible mechanism for how OM affects the force generation of myosin. Does this suggest that OM has potential therapeutic use for patients with this mutation? If so, it emphasizes the need to understand the molecular nature of the initial disruption to know which drugs may be useful to prevent disease progression.

Overall, this is an important paper with major implications for our understanding of HCM and the coupling of the ATPase and mechanical cycles in myosin. As always there are unanswered questions and the author could make some comments on these.

Revisions:

1) One mechanism by which a severe mutation can still result in HCM could be through a low expression level of the mutant proteins. Is there any evidence of loss of stability – or lower expression levels in C2C12 cells compared to WT?

2) The authors briefly discuss how this disruptive mutation could result in activation:

"Likewise, R712L-myosin could conceivably confer activating properties through the thin filament, or perhaps through activation of other myosins from the thick filament, although no mechanisms for those are apparent."

While it is not clear how this mutation could result in activation of the thin filament, the site of the mutation in the converter domain could conceivably disrupt the formation of the myosin interacting head motif by destabilizing the M.ADP.Pi pre-power stroke state. Again, the addition of OM could reverse this. The lower affinity of OM for the pre-powerstroke state is then the result of the binding energy required to stabilize the structure.

3) The apparent loss of the 2nd step in the laser trap assay (and its return in the presence of OM) brings up the question of the load dependence of the mutant myosin. Have the authors attempted to measure this in the motility assay or the laser trap?

4) While the Vmax of the ATPase remains unchanged for the mutant the KATPase value was ~ 4-fold smaller than WT. This seems unexpected in the context of the rest of the data presented. Similarly, OM does not change KATPase for the mutant but does appear to significantly increase KTF in the π release assay.

5) k-DAP and Vmax are of the same order for WT but k-DAP is 3 times Vmax for the mutant while KATPase and KTF are both very low. What is the predominant steady-state complex for the mutant – is it an actin attached step before π release? To phrase it a different way, how is π release accelerated >2 fold for the mutant while keeping Vmax the same?

6) Following on from 4 and 5 have the authors measured the duty ratio for the mutant?

---

## [Author Response]

Revisions:1) One mechanism by which a severe mutation can still result in HCM could be through a low expression level of the mutant proteins. Is there any evidence of loss of stability – or lower expression levels in C2C12 cells compared to WT?

Expression experiments show no differences in yield of recombinant protein that would suggest instability of R712L- or E497D-myosin when compared to WT-myosin (HMM fragments). We assessed 5 preparations each of WT- and R712L-myosin and analyzed the yield of purified myosin per unit culture dish for a fixed expression period, using a common infection protocol, comparable multiplicity of infection, and identical purification methods. Average yield of WT-myosin was 174 µg per p150 dish (range: 100 – 262) and 157 µg (range: 126 – 210) for R712L-myosin. SDS- PAGE analysis of each step from each purification from cell lysis through the final purified protein was performed to detect losses from individual steps. Comparable yields of the R712L proteins suggest no difference in the stability in this culture system. This information is now included in supplemental information (Figure 1—figure supplement 1A) and is discussed in the text.

2) The authors briefly discuss how this disruptive mutation could result in activation:"Likewise, R712L-myosin could conceivably confer activating properties through the thin filament, or perhaps through activation of other myosins from the thick filament, although no mechanisms for those are apparent."While it is not clear how this mutation could result in activation of the thin filament, the site of the mutation in the converter domain could conceivably disrupt the formation of the myosin interacting head motif by destabilizing the M.ADP.Pi pre-power stroke state. Again, the addition of OM could reverse this. The lower affinity of OM for the pre-powerstroke state is then the result of the binding energy required to stabilize the structure.

We agree that the mutation could reduce the population of the interacting-head structure on the thick filament, resulting in activation. We now include this possibility in the revised paper.

3) The apparent loss of the 2nd step in the laser trap assay (and its return in the presence of OM) brings up the question of the load dependence of the mutant myosin. Have the authors attempted to measure this in the motility assay or the laser trap?

We agree with the reviewers that the loss of the 2^nd^ step may affect the mechano-sensitivity of the myosin. However, measuring the load dependence of actin detachment is a substantial undertaking and is outside of the scope of the current work. We plan to further characterize this mutation in future studies, with load dependence being an experiment at the top of our priority list. The reviewer’s important point about possible effects of the loss of the second step on load sensitivity is now stated in the main text.

4) While the Vmax of the ATPase remains unchanged for the mutant the KATPase value was ~ 4-fold smaller than WT. This seems unexpected in the context of the rest of the data presented. Similarly, OM does not change KATPase for the mutant but does appear to significantly increase KTF in the π release assay.

We agree that it is interesting that the K_TF_ values for the mutant +/- OM are different, despite similar K_ATPase_ values. It is also surprising that the K_ATPase_ values for R712L are small. However, without knowing which rate constants define the steady-state parameters, it is difficult to give a definitive prediction about the relationship of the V_max,_K_ATPase_, and K_TF_ for the myosins +/- OM. What is surprising to the reviewer and to us, is that our results suggest the K_ATPase_ is not defined by the K_TF_, which one might predict if phosphate release is rate limiting, but this is not the case. Previous work by White et al. (Biochemistry, 1997, 36:11828) has suggested that the Vmax is defined by a combination of rate and equilibrium constants, including phosphate release, the equilibrium constant for ATP hydrolysis, and the affinities of the M.ATP and M.ADP.Pi states for actin. This point is now discussed in the text.

5) k-DAP and Vmax are of the same order for WT but k-DAP is 3 times Vmax for the mutant while KATPase and KTF are both very low. What is the predominant steady-state complex for the mutant – is it an actin attached step before π release? To phrase it a different way, how is π release accelerated >2 fold for the mutant while keeping Vmax the same?

Please see our response to #4. We do not know the rate limiting step for this myosin. In the case of R712L, the phosphate release step must not be limiting since its rate is accelerated without an overall increase in the rate of ATPase cycling. Additionally, exit from the strong binding states is also not rate limiting (Table 3). Future biochemical experiments that further explore ATP cleavage (e.g., quenched flow) and the transition from the weak-to-strong binding states are required. Although such experiments will help us further understand the effects of the R712L mutation, they will not change the primary conclusions of the current study.

6) Following on from 4 and 5 have the authors measured the duty ratio for the mutant?

The duty ratio is the fraction of the ATPase cycle time (determined by V_max_) that myosin occupies the strong-binding states, which we determined from the actin detachment rate in the optical trapping assay (*k*_1_ in Table 3). The calculated duty ratios in the absence of load are 0.095 for WT-myosin and 0.078 for R712L-myosin. We now include this information in the updated manuscript.